# Intraoperative Hypertension Is Associated with Postoperative Acute Kidney Injury after Laparoscopic Surgery

**DOI:** 10.3390/jpm13030541

**Published:** 2023-03-17

**Authors:** Yongzhong Tang, Bo Li, Wen Ouyang, Guiping Jiang, Hongjia Tang, Xing Liu

**Affiliations:** 1Department of Anesthesiology, The Third Xiangya Hospital, Central South University, Changsha 410013, China; 2Operation Center, The Third Xiangya Hospital, Central South University, Changsha 410013, China

**Keywords:** intraoperative hypertension, acute kidney injury, laparoscopic surgery, mean arterial pressure, exposure time

## Abstract

Background: It is well demonstrated that intraoperative blood pressure is associated with postoperative acute kidney injury (AKI); however, the association between severity and duration of abnormal intraoperative blood pressure (BP) with AKI in patients undergoing laparoscopic surgery remains unknown. Methods: This retrospective cohort study included 12,414 patients aged ≥ 18 years who underwent a single elective laparoscopic abdominal surgery during hospitalization between October 2011 and April 2017. Multivariate stepwise logistic regressions were applied to determine the correlation between the severity and duration of intraoperative mean arterial pressure (MAP, (systolic BP + 2 × diastolic BP)/3), acute intraoperative hypertension (IOTH) and postoperative AKI, in different periods of surgery. Results: A total of 482 hospitalized patients (3.9%) developed surgery-related AKI. Compared with those without IOTH or with preoperative mean MAP (80–85 mmHg), acute elevated IOTH (odds ratio, OR, 1.4, 95% CI, 1.1 to 1.7), mean MAP 95–100 mmHg (OR, 1.8; 95% CI, 1.3 to 2.7), MAP 100–105 mmHg (OR, 2.4; 95% CI, 1.6 to 3.8), and more than 105 mmHg (OR, 1.9; 95% CI, 1.1 to 3.3) were independent of other risk factors in a diverse cohort undergoing laparoscopic surgery. In addition, the risk of postoperative AKI appeared to result from long exposure (≥20 min) to IOTH (OR, 1.9; 95% CI, 1.5 to 2.5) and MAP ≥ 115 mmHg (OR, 2.2; 95% CI, 1.6 to 3.0). Intraoperative hypotension was not found to be associated with AKI in laparoscopic surgery patients. Conclusions: Postoperative AKI correlates positively with intraoperative hypertension in patients undergoing laparoscopic surgery. These findings provide an intraoperative evaluation criterion to predict the occurrence of postoperative AKI.

## 1. Introduction

Laparoscopic surgery is very popular because this technique is associated with reduced lesions, superior safety, shortened hospital stays, an early return to daily activity, and cosmetic acceptance of the operative scar [1,2]. Although uncommon, laparoscopy surgery was reported to increase the risk of acute kidney injury [3,4,5]. Acute kidney injury (AKI) is a serious postsurgical complication that occurs in a range of 3% to 40% [6,7], and it is a powerful predictor of various long-term adverse events [8] such as chronic kidney disease (during a median follow-up period of 3.3 years, approximately 50% of discharged patients who successfully recovered from hospital-acquired AKI were newly diagnosed with chronic kidney disease [9]), cardiovascular disease, and increased mortality (despite appropriate treatment, 12.32% of AKI patients died in the hospital, and 35% of AKI patients died within one year [10]). Unfortunately, there is no specific therapeutic regimen to solve this problem [11].

Many studies have shown that intraoperative blood pressure (BP), which provides organ perfusion, is a powerful prognostic marker for AKI. The overall evidence based on a diverse form of surgery shows that both hypotension and hypertension are associated with unfavorable outcomes [12], yet relatively little is known about how intraoperative BP affects surgery-related AKI in laparoscopic surgery patients. Therefore, we conducted a retrospective cohort study in elective laparoscopic surgery to evaluate the association between the severity and the duration of abnormal intraoperative BP and postoperative AKI, in different periods of surgery.

## 2. Materials and Methods

### 2.1. Data Sources

All de-identified data were obtained from the previously described structured hospital information system (HIS) and anesthesia information system of the Third Xiangya Hospital (Changsha, China) [13,14,15,16]. These systems could provide patient health information, such as information regarding enterprise master patient index (EMPI), diagnoses (International Classification of Disease (ICD-10) clinical diagnosis), symptoms, real-time vital signs, anesthetic drugs, and other perioperative data (preoperative drugs, laboratory tests, surgery date and sites, and so on). As a large tertiary hospital administered directly under the Chinese National Ministry of Education, the 3rd Xiangya hospital is one of the first thirteen hospitals awarded a four-star (maximum five-star) rating of the Grading Management of Health Information Interconnection and Standardization Maturity Evaluation in China [17]. This study complied with the ethical guidelines of the 1975 Declaration of Helsinki. The 3rd Xiangya Hospital institutional review board approved this study (No. R202233). Informed consent was not necessary, because all subjects of this retrospective study were anonymized. The reporting of this study conforms to the STROBE guidelines (Strengthening the Reporting of Observational Studies in Epidemiology) (Appendix A).

### 2.2. Subjects and Procedures

This study included adult patients (≥18 years) who underwent a single elective laparoscopic surgery during hospitalization between October 2011 and April 2017. Moreover, we excluded patients (*n* = 6646) with missing data (e.g., age, gender, intubation time, incision time, extubation time, or preoperative blood creatinine). Finally, 12,414 eligible patients were included.

The forms of laparoscopic surgery performed at our center include general abdominal, urology, and gynecological surgery. Laparoscopic gynecological surgery accounts for a large proportion (64.6%) in our hospital, which is reflected in our retrospective cohort (89.6% female patients). Moreover, 85.0% of patients who underwent laparoscopic surgery were of American Society of Anesthesiologists physical status (ASA) I–II, indicating that most of the patients in our study were healthy or complicated with mild systemic disease. Laparoscopic surgery has 4 phases: anesthesia induction, abdominal insufflation, abdominal de-sufflation, and emergence from general anesthesia. After entering the operating room and surgical safety check, all patients underwent general anesthesia induction with real-time monitoring. Artificial airway establishment such as ora-tracheal intubation was performed during the anesthesia induction. Pneumoperitoneum was established by a Veress needle and was maintained by continuous insufflation of CO_2_ with 11–13 mmHg intra-abdominal pressure. When the operation was over, we closed the pneumoperitoneum and began to perform recovery from anesthesia. When a patient’s consciousness and breathing returned, the ora-tracheal was removed and patients left the operating room. All patients had their endotracheal tubes removed after the surgery in our study. The patients’ intraoperative position could be the supine, lateral position, or reverse Trendelemburg position (common in gynecological or urological laparoscopic surgery) in this study.

### 2.3. Intraoperative BP

Intraoperative BP was recorded electronically using the anesthesia information system, from the time the patients entered the operating room until they returned to the wards. Blood pressure monitoring is divided into invasive and non-invasive BP. It was left up to the attending anesthesiologist to decide whether to conduct invasive blood pressure monitoring (according to the patient’s condition and the surgery risk). In this study, 28% of patients received invasive BP monitoring. Invasive arterial pressure was recorded every 30 s, and non-invasive BP was recorded every 5 min. If invasive and non-invasive BP measurements exist for the same time point, the intra-arterial BP was preferentially used. Mean arterial pressure (MAP) is widely used as an index for blood pressure, and it was calculated by the following formula: MAP = (systolic BP + 2 × diastolic BP)/3. Acute elevations in blood pressure (≥30% or ≥140/90 mmHg or MAP ≥ 105 mmHg) [18] during the operation were considered acute intraoperative hypertension (IOTH). Intraoperative hypotension (IOH) was defined as an intraoperative MAP decrease ≥ 40% and MAP < 70 mmHg at the same time, or isolated MAP < 60 mmHg [19]. The coefficient of variability (CV), equal to (standard deviation of intraoperative MAP)/(intraoperative mean MAP), was used to evaluate BP variability.

### 2.4. Outcomes

All included patients were followed up to discharge or within seven days after surgery, or until the development of surgery-related AKI. The surgery-related AKI was defined by the KDIGO (Kidney Disease: Improving Global Outcomes) AKI classification system [20]. If a serum creatinine increase ≥0.3 mg/dL in 48 h or a 1.5-fold increase in 7 days after surgery) using the peak-to-nadir serum creatinine difference. Briefly, the serum creatinine peak was the highest value during the first 48 h or 7 days after surgery, and the serum creatinine nadir was the lowest value recorded during the 7 days before surgery. The control group included all subjects who did not meet this criterion.

### 2.5. Confounders

Interested covariates were selected according to reported AKI risk factors [21] and our previous work [16]. These variables included age, gender, weight, preoperative BP (the average values before surgery within two weeks), eGFR (estimated glomerular filtration rate, calculated by the CKD-EPI (Chronic Kidney Disease Epidemiology Collaboration) creatinine equation [22]), ASA (American Society of Anesthesiologists) physical status, plasma biomarkers (e.g., routine blood tests, glucose, hepatic and renal function, and blood lipids), intraoperative heart rate, IOH, intraoperative CV of MAP, and final diagnoses (hypertension, COPD, diabetes mellitus, malignant tumor), as well as medications; for example, anesthetic drugs, intraoperative vasoactive agents, preoperative contrast media, preoperative lipid-lowering drugs, preoperative anti-diabetic drugs, and preoperative antihypertensive drugs. The total opioid use in the intraoperative period and recovery room was recorded, and values were converted to the equivalent alfentanil dose [20]. Less than 5% of items per variable were missing, and the data in this study had a high rate of integrity. Missing data were imputed using an average value (if continuous data) or multiple imputations (if category data) [23].

### 2.6. Statistical Analysis

The included patients were divided into 9 groups according to intraoperative mean MAP: <70 mmHg, 70–75 mmHg, 75–80 mmHg, 80–85 mmHg, 85–90 mmHg, 90–95 mmHg, 95–100 mmHg, 100–105 mmHg, and ≥105 mmHg. We also divided all patients into IOTH or non-IOTH, as well as IOH or non-IOH. In subgroup analysis, the surgery was divided into three periods, anesthesia induction (time between intubation and skin incision), after skin incision (specifically, 30 min after skin incision), and during emergence from general anesthesia (the time between completion of surgery and extubation). We calculated the total number of minutes spent with IOTH, with a MAP of more than 105 mmHg, more than 110 mmHg, and more than 115 mmHg for each case. For intraoperative hypertension, patients were categorized as having spent 0, 1–5, 6–10, 11–20, or more than 20 min (Table 1).

By using SAS (version 9.4) and IBM SPSS Version 22.0 (SPSS Inc., Chicago, IL, USA), all of the analyses and calculations were performed. Means ± standard deviations (SDs) were presented for continuous variables, the median and interquartile range (IQR) were expressed for skewed distribution data, and proportions were presented for categorical variables. Student’s *t*-tests or Mann–Whitney U tests were used to compare differences between continuous variables. A *p*-value of 0.05 (2-tailed) was regarded as statistical significance.

The sample size was not calculated before analyzing this data. All covariates (the variables that are shown in Table 2) were included for analysis by single-factor correlation to identify factors associated with postoperative AKI, and less significant (*p* ≥ 0.05) clinical or statistical variables were excluded. In the final model, if variation inflation factors (VIF) ≥ 5, multicollinearity was determined to be presented.

Stepwise logistic regression models were used to estimate the association of postoperative acute kidney injury with the severity and duration of intraoperative BP. In addition, the association between postoperative AKI and intraoperative hypertension in different periods of surgery was estimated, including throughout the operation, during anesthesia induction, after skin incision, and during emergence from general anesthesia. Finally, we explored the association between the exposure time of intraoperative hypertension and AKI, in different periods of surgery. We adjusted the models for the following: age, gender, ASA physical status, complicated with diabetes, preoperative usage of β-Receptor antagonists, CCBs, lipid-lowering drugs, intraoperative usage of cisatracuram and vasoactive drugs, preoperative eGFR, urea nitrogen, intraoperative mean heart rate, duration of surgery, duration of intubation, and whether or not a patient entered the intensive care unit after surgery. We used ten-fold cross-validation and receiver operating characteristic (ROC) to assess the predictive accuracy of the model.

We conducted interaction and sensitivity analyses to assess the robustness of our findings. First, to explore whether the risk of AKI for exposure to intraoperative hypertension was influenced by IOH, we analyzed the joint effects of IOH and IOTH. Second, we explored whether the risk of AKI was driven by preoperative comorbidities, the multivariable analysis was repeated in subgroups of ASA grade I-II, without hypertension/diabetes/cancer, and age < 65 years patients, respectively. Third, we restricted patients with intraoperative blood loss of less than 1000 mL, and without preoperative anemia (preoperative hemoglobin less than 90 g/L). Finally, we re-evaluated the association of AKI with intraoperative BP in patients with/without invasive BP monitoring.

## 3. Results

After screening and application of selection criteria, 12,414 patients were enrolled (Figure 1). There were 482 patients (3.9%) who developed surgery-related AKI in our study. The baseline characteristics of this study are shown and compared among the defined intraoperative MAP thresholds in Table 1. The mean age was 44.2 ± 14.1 years, and 89.5% of the patients were female. Overall, 85.0% of patients who underwent laparoscopic surgery were ASA I-II, indicating that most of the patients in our study were healthy or complicated with mild systemic disease. The median length of postoperative hospital stay was 5 days and the study population was largely comparable to the excluded patients (Appendix A).

Figure 2 shows selected characteristics of intraoperative BP in patients who underwent elective laparoscopic surgery, among which 46.6% developed acute intraoperative hypertension during whole surgery, 39.1% developed IOTH after anesthesia induction, and 34.9% developed IOTH after skin incision (Figure 2A). A substantial fraction of all hypertension happened during the time between anesthesia induction and skin incision. A total of 56.8% of the elective laparoscopic surgery patients in the study had MAP values ranging from 70 to 85 mmHg (Figure 2C). The intraoperative mean MAP tended to decrease as the duration of anesthesia progressed, while this value tended to increase after the skin incision (Figure 2D). Patients who developed AKI had a higher incidence of acute intraoperative hypertension, a higher MAP mean level (83.4 ± 8.7 vs. 87.7 ± 10.3, *p* < 0.001), and a longer duration of abnormally elevated blood pressure (*p* < 0.001) compared to non-AKI patients; these trends were detected for all periods of surgery. A total of 34.4% of patients developed intraoperative hypotension in our study (Figure 2B), and there was no statistically significant difference in MAP variability or the incidence of IOH between the two groups after adjusting for confounders.

A binary logistic regression analysis revealed an independent relationship between intraoperative hypertension and postoperative AKI. Compared to those without acute intraoperative hypertension, the adjusted OR (odds ratio) of AKI was 1.4 (95% CI, 1.1 to 1.7) in patients with acute intraoperative hypertension (Figure 3A), with ORs of AKI of 1.28 during anesthesia induction and 1.23 during skin incision.

Compared with the preoperative intraoperative mean MAP level (80–85 mmHg), a MAP of 95–100 mmHg (OR, 1.8; 95% CI, 1.3 to 2.7), 100–105 mmHg (OR, 2.4; 95% CI, 1.6 to 3.8), or more than 105 mmHg (OR, 1.9; 95% CI, 1.1 to 3.3) led to AKI in an independent and graded manner (Figure 3B); the ORs of AKI were 2.9 and 1.8 for patients with MAP ≥ 105 mmHg during the anesthesia induction and during skin incision, respectively (Figure 3C,D).

With increasing durations of intraoperative hypertension, there was a trend toward an increased risk of postoperative AKI. Compared with patients who spent no time with intraoperative hypertension, the patients with the longest IOTH durations (more than 20 min) had a 1.6-fold increased risk of AKI (Figure 4A). Moreover, this trend of increased risk was exacerbated with increasing intraoperative MAP: with the highest risk (2.2-fold increase) observed for the patient that had ≥115 mmHg values for more than 20 min (Figure 4B). There were no interactions between IOTH and IOH. These relationships were preserved qualitatively in interaction analyses and sensitivity analyses (Appendix A and Appendix A).

## 4. Discussion

This retrospective study evaluated associations between the degree and duration of elevated intraoperative BP with surgery-related AKI in 12,414 elective laparoscopic surgery patients at our center. The main findings of this study are as follows: (1) the incidence of postoperative AKI in elective laparoscopic surgery patients was 3.9%; (2) acute elevation of intraoperative blood pressure and mean MAP ≥ 95 mmHg were independent of other risk factors in elective laparoscopic surgery patients; (3) for all periods of surgery, as well as for the anesthesia induction and skin incision periods specifically, intraoperative hypertension was significantly associated with AKI; (4) prolonged exposure (≥20 min) to intraoperative hypertension elevated the risk of postoperative AKI by 1.6-fold, with the highest risk (2.2-fold increase) detected for the exposure to intraoperative MAP ≥ 115 mmHg patients.

We confirmed that elevated intraoperative BP is an independent risk factor for postoperative AKI in a diverse cohort of patients undergoing laparoscopic surgery. Our results extend previous work [24,25,26,27] by showing that the severity and duration of intraoperative hypertension are significantly associated with AKI. One strength of this study was that acute kidney injury could be found through the structured hospital information system of the 3rd Xiangya Hospital; the hospital information system serves as the only portal for clinical data for all patients and provides reliable and quality-controlled data. Thus, a huge number of blood pressure measurements can be electronically recorded on a per-minute basis, and blood pressure episodes can be defined reliably. Another strength of our study was the large sample size with stable multivariable modeling. The cross-validation of the major model of IOTH, MAP level, and AKI showed a statistically significant predictive accuracy for AKI (AUC 0.79; 95% CI, 0.76–0.80; AUC 0.79; 95% CI, 0.77–0.81). Third, we included predictor variables that were available from the hospital information system, including preoperative BP and many other risk factors that could affect intraoperative BP and postoperative AKI. In addition, our results are consistent with limited data from a randomized clinical study including elderly patients who underwent gastrointestinal surgery [28]; in that prospective randomized study that included 678 elderly patients, a high intraoperative MAP level (96–110 mmHg) increased the risk of surgery-related AKI. In vasopressor-dependent cardiovascular surgery, patients with AKI progression had greater diastolic pressure, mean perfusion pressure, and diastolic perfusion pressure compared to non-AKI patients [29]. It is noteworthy that unlike other studies [24,25,26,27] that emphasized the association between intraoperative hypotension and AKI, we cannot obtain this conclusion. Maybe because 85% of the included subjects in our study were ASA I-II, and few of them had a sustained period of intraoperative MAP less than 65 mmHg.

Considering the effect of surgery stimulation on perioperative blood pressure and postoperative AKI, we restricted the type of elective laparoscopic surgery in this study. Intraoperative hypertension is very common in laparoscopic surgery. A total of 46.6% developed acute intraoperative hypertension in our study. Moreover, for the anesthesia induction and skin incision periods specifically, as well as for all periods of surgery, intraoperative hypertension was significantly associated with AKI. These results are closely related to the particularity of laparoscopic surgery. Laparoscopic surgery has four phases: anesthesia induction, abdominal insufflation, abdominal desufflation, and emergence from general anesthesia. With the beginning of laparoscopic surgery, intraabdominal pressure increases followed by the carbon dioxide artificial pneumoperitoneum, as does the intraoperative blood pressure. Usually, the sympathetic activation of general anesthesia, intubation, pneumoperitoneum, and surgical stimulation can lead to a 20–30 mmHg increase in normotensive individuals’ blood pressure [30], and this situation will be more serious in high-risk patients because perioperative autonomic nerve function is significantly impaired in hypertensive and elderly patients [31,32]. Moreover, the reverse Trendelemburg position in laparoscopic surgery could also lead to hemodynamic changes (e.g., increased BP and systemic vascular resistance, decreased cardiac index) [33]. A clinical study included 12 men of ASA I–II undergoing robot-assisted laparoscopic prostatectomy reported that left ventricular ejection fraction (decreased 7%), systolic blood pressure (increased 12 mmHg), and diastolic blood pressure (increased 11 mmHg) changed significantly after head down tilting and pneumoperitoneum establishment [34].

Our findings also suggest that anesthesiologists should seek to avoid long-term exposure to intraoperative hypertension in laparoscopic surgery. Elevated intraoperative MAP offers an informative reflection of the patient’s general condition [35] (e.g., complicated diseases, hemorrhage, sepsis, blood volume, cardiac output), surgical stress, intubation stimulation, anesthesia depth, intra-abdominal pressure, and CO_2_ inflation time. Although AKI is usually multifactorial, hemodynamic instability, hypoxia/ischemia, sepsis, and drug toxicity are commonly implicated [36]. Unlike open surgery, the key process of laparoscopic surgery, i.e., increased intra-abdominal pressure (IAP) followed by carbon dioxide (CO_2_) artificial pneumoperitoneum, has a great influence on hemodynamics and renal physiology [37]. Elevated IAP can increase mean arterial pressure (MAP), activate the RAAS (renin-angiotensin-aldosterone system) [38] and lead to direct compression of the inferior vena cava, aortic and renal vascular, and lead to a reduction in renal blood flow [39]. An animal experiment [40] showed that in well-hydrated pigs, superficial renal cortical blood flow was reduced by 60% after 2 h of CO_2_ insufflation, and the renal blood flow recovered to its pre-insufflated state after desufflation. In clinical settings, renal functions might be affected by direct compression of the renal vascular and kidneys, general anesthesia, and operative procedures of laparoscopic surgery. In a previous study (*n* = 14) [41], compared with the Low-IAP group (4 mmHg), significant decreases in urine output and glomerular filtration rate were observed in the High-IAP group (12 mmHg) after the first 30–60 min of pneumoperitoneum. A single-center prospective cohort study (*n* = 64) [42] reported that the exposure of CO_2_ inflation time and IAP is an impact factor of laparoscopic abdominal postoperative AKI.

This study has several limitations. First, we chose patients who underwent elective laparoscopic surgery. A total of 85.0% of patients were ASA I-II, and few had a sustained period of intraoperative MAP < 65 mmHg. Additionally, the incidence of postoperative AKI was 3.9%, which was lower than the values for other types of surgery (e.g., cardiac surgery, major noncardiac surgery, and so on) [6,43]. Surgery patients with good healthy and high organ-specific physiological reserve may not have any substantial postoperative organ dysfunction, even high-risk surgeries with acute stresses, such as hemodynamic instability and prolonged ischemia/hyperfusion of major organs. As a result, although the restricted surgery type enhanced the internal validity of this study, our results are likely to represent the risk for patients in good condition; these findings could not generalize to other kinds of surgery, emergency surgery, or different methods of anesthesiology, including surgery in which the patient has low intraoperative BP. Our study was unfortunately also limited by a lack of monitoring of the depth of anesthesia and renal vascular tone and blood flow. The relationship between IAP, MAP, renal perfusion, and AKI should be further explored.

## 5. Conclusions

In conclusion, among patients undergoing elective laparoscopic surgery, acute intraoperative hypertension, a mean intraoperative MAP greater than 95 mmHg, and exposure to 20 min intraoperative hypertension are independently associated with an increased risk of postoperative AKI. These findings provide a motivation for the development of intraoperative management strategies for the prevention of postoperative AKI because, unlike other risk factors for AKI, intraoperative physiology can usually be controlled, especially in general anesthesiology patients.

## Figures and Tables

**Figure 1 jpm-13-00541-f001:**
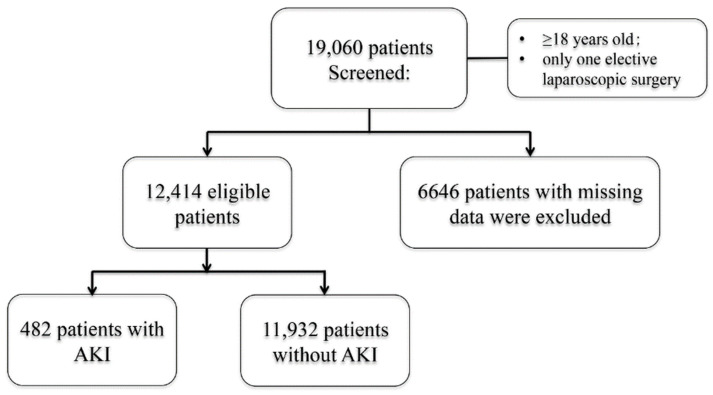
Flow chart of patient selection.

**Figure 2 jpm-13-00541-f002:**
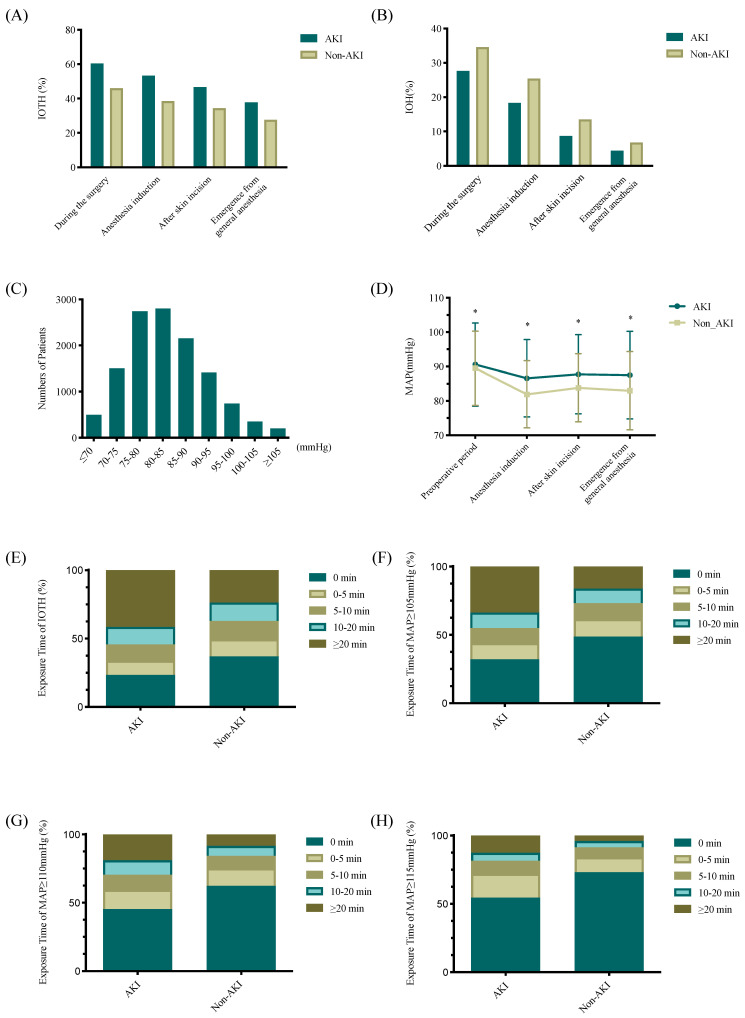
Characteristics of intraoperative blood pressure in the patients who underwent laparoscopic surgery: (**A**) The incidence of acute intraoperative hypertension in patients with postoperative AKI and without AKI, recorded during the surgery, upon anesthesia induction, after skin incision, and at emergence from general anesthesia; (**B**) The incidence of acute intraoperative hypotension in patients with postoperative AKI and without AKI, recorded during the surgery, upon anesthesia induction, after skin incision, and at emergence from general anesthesia; (**C**) Histogram of intraoperative MAP values; (**D**) Intraoperative MAP values (mean ± standard deviation) in patients with postoperative AKI and without AKI, recorded before the surgery, upon anesthesia induction, after skin incision, and at emergence from general anesthesia. * Compared with non-AKI, *p* < 0.05; (**E**) Distribution for the duration of acute intraoperative hypertension in patients with postoperative AKI and without AKI; (**F**) Distribution for the duration of intraoperative MAP ≥ 105 mmHg in patients with postoperative AKI and without AKI; (**G**) Distribution for the duration of intraoperative MAP ≥ 110 mmHg in patients with postoperative AKI and without AKI; (**H**) Distribution for the duration of intraoperative MAP ≥ 115 mmHg in patients with postoperative AKI and without AKI.

**Figure 3 jpm-13-00541-f003:**
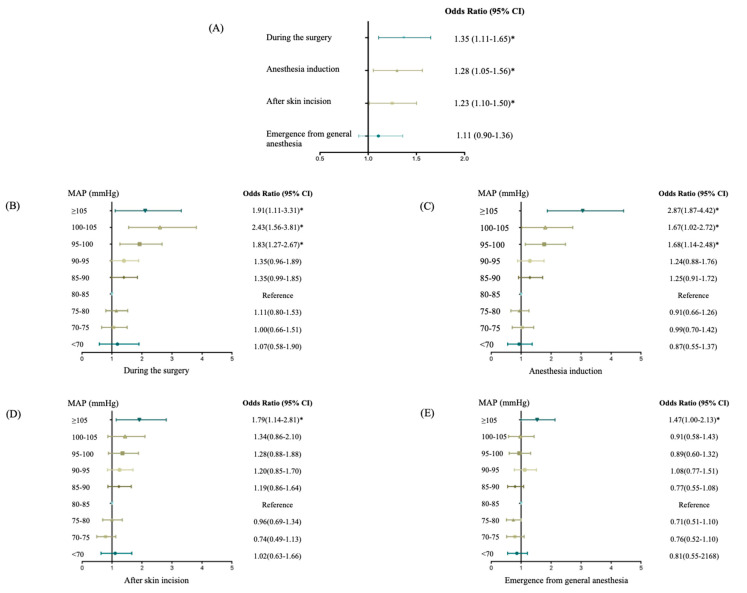
Forest plots of adjusted odds ratios for the association between postoperative AKI and intraoperative hypertension during different periods of laparoscopic surgery: (**A**) The adjusted odds ratio for an association between postoperative AKI and IOTH during the indicated periods of the surgery; (**B**) The adjusted odds ratio for an association between postoperative AKI and mean MAP levels during surgery (all periods); (**C**) The adjusted odds ratio for an association between postoperative AKI and mean MAP levels during the anesthesia induction period; (**D**) The adjusted odds ratio for an association between postoperative AKI and mean MAP levels during the skin incision period; (**E**) The adjusted odds ratio for an association between postoperative AKI and mean MAP levels in the emergence from general anesthesia period. * *p* < 0.05 compared with the reference value (MAP 80–85 mmHg).

**Figure 4 jpm-13-00541-f004:**
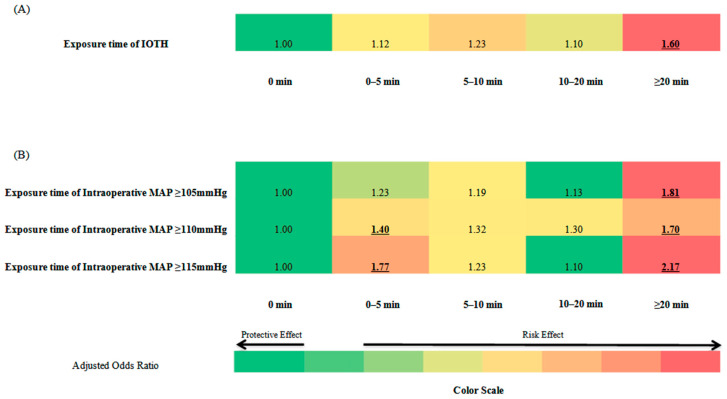
Heatmap of the adjusted odds ratio for the association between postoperative AKI and exposure time of intraoperative hypertension: (**A**) The adjusted odds ratio for an association between postoperative AKI and the exposure time of acute intraoperative hypertension; (**B**) The adjusted odds ratio for an association between postoperative AKI and the duration for which MAP ≥ 105 mmHg, MAP ≥ 110 mmHg, or MAP ≥ 115 mmHg. Underlined numbers, *p* < 0.05 compared with the reference.

**Table 1 jpm-13-00541-t001:** The definition of intraoperative blood pressure and the study group of patients who underwent elective laparoscopic surgery.

	MAP	IOTH	IOH
Definition	MAP = (SBP + 2 × DBP)/3, where SBP is systolic blood pressure and DBP is diastolic blood pressure	Acute intraoperative hypertension. Acute elevations in blood pressure (≥30% or ≥140/90 mmHg or MAP ≥ 105 mmHg) during the operation.	Acute intraoperative hypotension. The intraoperative MAP decreases by more than 40% with MAP < 70 mmHg at the same time, or an isolated MAP < 60 mmHg.
Groups	The entire study population was divided into 9 groups based on intraoperative mean MAP: <70 mmHg, 70–75 mmHg, 75–80 mmHg, 80–85 mmHg, 85–90 mmHg, 90–95 mmHg, 95–100 mmHg, 100–105 mmHg, and ≥105 mmHg.	All patients were divided into two groups based on whether IOTH occurred.	All patients were divided into two groups based on whether IOH occurred.
Subgroups	1. According to different periods of surgery, we explored associations between intraoperative blood pressure and AKI during anesthesia induction (i.e., the time between intubation and skin incision), after skin incision (specifically, 30 min after skin incision), and during emergence from general anesthesia (the time between completion of surgery and extubation).		
	2. We categorized patients as having spent 0, 1–5, 6–10, 11–20, or more than 20 min of cumulative IOTH time, a MAP of more than 105, more than 110, more than 115. Associations between intraoperative hypertension and AKI were estimated for the different exposure times in the aforementioned different periods of surgery.		

DBP, diastolic blood pressure; MAP, mean arterial pressure; IOH, intraoperative hypotension; IOTH, intraoperative hypertension; SBP, systolic blood pressure.

**Table 2 jpm-13-00541-t002:** Baseline characteristics of patients who underwent elective laparoscopic surgery, categorized by whether postoperative AKI was diagnosed.

	Postoperative Acute Kidney Injury		Total
	Yes (*n* = 482)	No (*n* = 11,932)	
Age (years) *	48.7 ± 13.8	44.0 ± 14.0	44.2 ± 14.1
Male sex (%) *	38.8	9.2	10.4
Weight (kg)	56.8 ± 10.0	56.8 ± 9.5	56.8 ± 9.5
Preoperative SBP (mmHg) *	121.0 ± 17.5	118.4 ± 15.5	118.5 ± 15.6
Preoperative DBP (mmHg) *	75.4 ± 10.8	75.1 ± 9.8	75.1 ± 9.8
Preoperative MAP (mmHg) *	90.6 ± 12.1	89.5 ± 10.8	89.6 ± 10.9
eGFR (%) *			
<30 mL/min/1.73 m^2^	2.3	0.6	0.6
30–60 mL/min/1.73 m^2^	2.9	4.1	4.1
≥60 mL/min/1.73 m^2^	94.8	95.3	95.3
ASA physical status (%) *			
I	23.4	16.2	16.5
II	59.3	68.9	68.5
III	16.8	14.2	14.3
IV	0.4	0.7	0.7
Intraoperative vital signs			
Mean SBP (mmHg) *	119.1 ± 14.1	113.1 ± 11.7	113.3 ± 11.9
Mean DBP (mmHg) *	72.0 ± 9.5	68.5 ± 8.1	68.6 ± 8.1
Mean MAP (mmHg) *	87.7 ± 10.3	83.4 ± 8.7	83.5 ± 8.8
CV of MAP	12.3 ± 3.3	12.1 ± 3.3	12.1 ± 3.3
IOTH (*n*, %) *	291, 60.4%	5491, 46.0%	5782, 46.6%
IOH (*n*, %) *	133, 27.6%	4134, 34.6%	4267, 34.4%
Heart rate (beats/minute) *	72.1 ± 10.1	70.2 ± 8.9	70.3 ± 9.0
Preoperative complications			
History of hypertension (%) *	18.3	11.6	11.8
Presence of diabetes (%) *	10.8	5.0	5.2
Presence of COPD (%)	0.6	1.1	1.1
Presence of malignant tumor (%) *	6.2	2.1	2.3
Preoperative medications			
CCBs (%) *	8.5	2.2	2.6
RASIs (%) *	0.8	0.3	0.3
β-Receptor antagonists (%) *	1.5	0.2	0.3
α-Receptor antagonists (%) *	1.2	0.2	0.3
Diuretics (%)	0.0	0.1	0.1
Oral antidiabetic drugs (%) *	1.0	0.2	0.3
Insulin (%) *	5.6	2.8	2.9
Contrast drugs (%) *	1.5	0.4	0.5
Lipid-lowering drugs (%) *	1.9	0.3	0.3
Preoperative laboratory tests			
Neutrophils (*10^9^/L)	4.3 ± 2.5	4.1 ± 2.3	4.1 ± 2.3
Lymphocytes (*10^9^/L)	1.8 ± 0.7	1.8 ± 0.7	1.8 ± 0.7
Red blood cells (*10^12^/L) *	4.1 ± 0.6	4.2 ± 0.5	4.2 ± 0.5
Hemoglobin (g/L)*	120.1 ± 20.6	121.8 ± 18.5	121.7 ± 18.6
Blood platelets (*10^9^/L)	223.8 ± 75.9	230.3 ± 76.8	230.1 ± 76.8
Glucose (mmol/L) *	5.2 ± 1.9	4.9 ± 1.2	4.9 ± 1.3
Uric acid (µmol/L) *	264.3 ± 95.6	260.9 ± 80.6	261.0 ± 81.2
BUN (mmol/L) *	4.8 ± 2.9	4.4 ± 1.6	4.4 ± 1.7
Albumin (g/L) *	40.6 ± 4.9	41.4 ± 4.2	41.4 ± 4.3
LDL-C (mmol/L)	2.5 ± 0.6	2.5 ± 0.5	2.5 ± 0.5
Total cholesterol (mmol/L)	4.6 ± 0.8	4.6 ± 0.7	4.6 ± 0.7
ALT (u/L) *	17.0 (12.8, 27.3)	16.0 (12.0, 24.0)	16.0 (12.0, 24.0)
AST (u/L)	19.0 (17.0, 26.0)	19.0 (16.0, 23.0)	19.0 (16.0, 23.0)
Intraoperative medications			
Midazolam (mg)	2.0 (1.0, 3.0)	2.0 (2.0, 3.0)	2.0 (2.0, 3.0)
Anesthetic-inducing alfentanils ^a^ (mg)	1.5 (0.5, 2.0)	1.0 (0.0, 2.5)	1.0 (0.0, 2.5)
Sevoflurane (mL)	15.0 (5.8, 20.0)	15.0 (5.0, 20.0)	15.0 (5.0, 20.0)
Propofol (mL)	48.1 (35.0, 50.0)	48.1 (30.0, 50.0)	48.1 (30.0, 50.0)
Alfentanils ^a^ (mg) *	17.0 (14.4, 28.0)	15.0 (13.5, 19.0)	15.0 (13.5, 20.0)
NSAIDs (%)	50.2	52.4	52.3
Dexmedetomidine (%)	6.0	7.8	7.7
Rocuronium (mg)	0.0 (0.0, 40.0)	0.0 (0.0, 40.0)	0.0 (0.0, 40.0)
Cisatracurium (mg) *	7.5 (0.0, 13.0)	5.0 (0.0, 10.0)	5.0 (0.0, 10.0)
Vasoactive drugs (%) *	14.9	7.1	7.4
Blood transfusion (%) *	17.0	9.7	10.0
Transfusion volume (mL) *	2200 (1600, 3100)	1600 (1300, 2500)	1600 (1450, 2600)
Other intraoperative information			
Invasive BP monitoring (%) *	43.8	27.4	28.0
Intraoperative blood loss ≥ 1000 mL (%) *	6.2	2.6	2.7
Operation time (hours) *	2.2 (1.4, 3.2)	1.7 (1.1, 2.5)	1.7 (1.1, 2.6)
Duration of intubation (hours) *	3.3 (2.2, 4.3)	2.4 (1.8, 3.4)	2.4 (1.8, 3.4)
Postoperative hospital stay (days) *	8.0 (5.0, 11.0)	5.0 (4.0, 9.0)	5.0 (4.0, 9.0)
The rate of converting to open abdominal surgery (%) *	4.4	1.3	1.4
Transfer to ICU after surgery (%) *	3.3	0.7	0.8
Type of surgery (%) *			
General abdominal surgery	45.9	29.8	30.4
Urology surgery	14.3	4.6	4.9
Gynecological surgery	39.8	65.6	64.6

ALT, glutamic pyruvic transaminase; ASA, American Society of Anesthesiologists; AKI, acute kidney injury; AST, aspartate aminotransferase; BUN, blood urea nitrogen; CCB, calcium channel blocker; COPD, chronic obstructive pulmonary disease; CV, coefficient of variability; DBP, diastolic blood pressure; LDL-C, low-density lipoprotein cholesterol; MAP, mean arterial pressure; ICU, intensive care unit; IOH, intraoperative hypotension; IOTH, intraoperative hypertension; NSAIDs, nonsteroidal anti-inflammatory drugs; SBP, systolic blood pressure; RASIs, renin-angiotensin system inhibitors. ^a^ The total opioid use was converted to the equivalent alfentanil dose. Values are the mean ± SD, number proportion, or median (IQR [range]) if the data exhibited a skewed distribution; comparisons were performed using chi-squared tests for the categorical variables; Student’s *t*-tests were used for normally distributed continuous variables, and Kruskal-Wallis tests were used for non-normally distributed continuous variables (Mann-Whitney U test between two groups). * *p* < 0.05.

## Data Availability

Detailed information about the datasets used and/or analyzed during the current study can be obtained from the corresponding author.

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
