# Peer review of "Intraoperative Hypertension Is Associated with Postoperative Acute Kidney Injury after Laparoscopic Surgery"

_jpm, 2023, doi:10.3390/jpm13030541_

Round 1
Reviewer 1 Report
This is a well-conceived and produced study of postoperative acute kidney injury.
Specific comments:
In the abstract MAP should be defined.
At the end of the introduction, abnormal intraoperative BP should be defined.
On page 8, in the outcomes section, the point at which a patient did not meet AKI was not specified. Please specify what constitutes a 'control'.
Page 19, the label for Figure 2B is missing a time point.
Page 20, second paragraph, I think you mean 'logistic regression' and not 'linear regression'.
Author Response
Reviewer1:
This is a well-conceived and produced study of postoperative acute kidney injury.
Specific comments:
In the abstract MAP should be defined.
Thanks for you advices. We have added the definition in abstract(Line 51-52).
At the end of the introduction, abnormal intraoperative BP should be defined.
Thanks for you advices. We have added the definition in introduction.
On page 8, in the outcomes section, the point at which a patient did not meet AKI was not specified. Please specify what constitutes a 'control'.
Thanks for you advices. We have specified the definition of control group.
Page 19, the label for Figure 2B is missing a time point.
Thanks for you advices. We have updated the time point in Figure 2B.
Page 20, second paragraph, I think you mean 'logistic regression' and not 'linear regression'.
Thanks for you advices. We have updated the sentences in page 20.
The abstract is very well structured, concise, but probably a bit long, considering that, as a rule, it is recommended that the abstract not exceed 200 words. In this case, the authors declare that the abstract contains 255 words.
Thanks for you advices. We have shorted the abstract.
The 5 keywords are well chosen and suggestive for the manuscript.
On a scale of 1 to 10, I will give 9 points for the abstract.

Reviewer 2 Report
Abstract:
The abstract is very well structured, concise, but probably a bit long, considering that, as a rule, it is recommended that the abstract not exceed 200 words. In this case, the authors declare that the abstract contains 255 words.
The 5 keywords are well chosen and suggestive for the manuscript.
On a scale of 1 to 10, I will give 9 points for the abstract.
Introduction:
The introduction, which has less than one page allocated, is too short. The theme of the manuscript seems presented in a telegraphic way, even if the text is supported by 12 bibliographic references.
On a scale of 1 to 10, I agree 8 points for introduction.
Methodology:
This is an exceptional chapter, which describes, in all details (in text and in the 2 tables, which are welcome in this chapter), how the study was carried out, how the data were collected, selected, and analyzed. The large number of patients included in the study (12.414) allows a correct statistical analysis of the data obtained. The methodology deserves maximum marks, without discussion.
On a scale of 1 to 10, I agree 10 points for methodology.
Results:
The results are well presented, clearly, in the text and 4 figures, totaling approximately 6 pages. I appreciate the large number of cases included in the study; this fact gives strength to the results. I have no suggestions for improvement for this chapter.
On a scale of 1 to 10, I agree 10 points for results.
Discussion:
The discussion chapter is well written, supported by sufficient bibliographic references. The authors know how to value their work well, they start the discussions by emphasizing the 3 strengths of the study carried out (in their view), continue by putting their results in the context of data from the literature and conclude by showing the limitations of their study. It is an intelligent way of presenting, with which I find no flaws worth mentioning.
In this situation, on a scale of 1 to 10, I agree 10 points for discussion.
Conclusion:
The conclusions are correctly written, they conclude this manuscript in a good manner, without forgetting to indicate potential future research directions. The only observation, minor by the way, is that the conclusions chapter should be highlighted as such.
On a scale of 1 to 10, I agree 9 points for conclusions.
Bibliography/References:
43 references, current, correctly written and correctly quoted in the text, I consider to be sufficient for this manuscript.
On a scale of 1 to 10, I agree 9 points for the bibliography.
Figures/Tables:
I identified 4 figures and 2 tables, of good quality, with satisfactory resolution, which are necessary and useful for the manuscript.
On a scale of 1 to 10, I agree 9 points for this chapter.
Review Decision:
Accept after minor revision.
Author Response
Reviewer2:
Introduction:
The introduction, which has less than one page allocated, is too short. The theme of the manuscript seems presented in a telegraphic way, even if the text is supported by 12 bibliographic references.
On a scale of 1 to 10, I agree 8 points for introduction.
Thanks for your suggestions. We have added the words of introduction.
Methodology:
This is an exceptional chapter, which describes, in all details (in text and in the 2 tables, which are welcome in this chapter), how the study was carried out, how the data were collected, selected, and analyzed. The large number of patients included in the study (12.414) allows a correct statistical analysis of the data obtained. The methodology deserves maximum marks, without discussion.
On a scale of 1 to 10, I agree 10 points for methodology.
Thanks for your 10 points.
Results:
The results are well presented, clearly, in the text and 4 figures, totaling approximately 6 pages. I appreciate the large number of cases included in the study; this fact gives strength to the results. I have no suggestions for improvement for this chapter.
On a scale of 1 to 10, I agree 10 points for results.
Thanks for your 10 points.
Discussion:
The discussion chapter is well written, supported by sufficient bibliographic references. The authors know how to value their work well, they start the discussions by emphasizing the 3 strengths of the study carried out (in their view), continue by putting their results in the context of data from the literature and conclude by showing the limitations of their study. It is an intelligent way of presenting, with which I find no flaws worth mentioning.
In this situation, on a scale of 1 to 10, I agree 10 points for discussion.
Thanks for your 10 points.
Conclusion:
The conclusions are correctly written, they conclude this manuscript in a good manner, without forgetting to indicate potential future research directions. The only observation, minor by the way, is that the conclusions chapter should be highlighted as such.
On a scale of 1 to 10, I agree 9 points for conclusions.
Thanks for your 9 points. We have updated a bit of the conclusion.
Bibliography/References:
43 references, current, correctly written and correctly quoted in the text, I consider to be sufficient for this manuscript.
On a scale of 1 to 10, I agree 9 points for the bibliography.
Thanks for your 9 points.
Figures/Tables:
I identified 4 figures and 2 tables, of good quality, with satisfactory resolution, which are necessary and useful for the manuscript.
On a scale of 1 to 10, I agree 9 points for this chapter.
Thanks for your 9 points.
Review Decision:
Accept after minor revision.
